



# A Flexible ROMS-based Hybrid Coupled Model for ENSO Studies– Model Formulation and Performance Evaluation

Yang Yu[1,2], Yin-nan Li[1], Rong-Hua Zhang[3,4], *, Shu-Hua Chen[2], Yu-Heng Tseng[5], Wenzhe Zhang[1], Hongna Wang[1,4]

[1] Key Laboratory of Ocean Circulation and Waves, Institute of Oceanology, Chinese Academy of Sciences, Qingdao, China
[2] University of California-Davis, Davis, CA, USA
[3] School of Marine Sciences, Nanjing University of Information Science and Technology, Nanjing, China
[4] Laoshan Laboratory, Qingdao 266237, China
[5] Institute of Oceanography, National Taiwan University, Taipei, Taiwan

*Correspondence to*: Rong-Hua Zhang (Rong-Hua Zhang, rzhang@nuist.edu.cn)

**Abstract.** The El Niño and Southern Oscillation (ENSO) constitutes the most prominent interannual climate variation mode in the climate system, originating from ocean-atmosphere interactions in the tropical Pacific. Accurately modeling ENSO variation has consistently posed a great challenge, exhibiting strong model-dependent representations and simulations of ENSO. This study presents a novel Hybrid Coupled Model (HCM), denoted as HCM$_{ROMS}$, built upon the Regional Ocean Modeling System (ROMS) that has been widely used for regional modeling studies. For basin-wide applications to the tropical Pacific, here, the ROMS is incorporated with a statistical atmospheric model, which is based on singular value decomposition (SVD), capturing interannual relationships of atmospheric perturbations such as wind stress and freshwater flux anomalies with sea surface temperature (SST) anomalies. The model is constructed in a flexible way so that various components representing atmospheric forcing and oceanic biogeochemistry can be easily included as a module in the HCM$_{ROMS}$. Results demonstrate that the HCM$_{ROMS}$ can simulate a stable quasi-three-year ENSO cycle when the interannual wind stress coupling coefficient, $\alpha_\tau$, is set at 1.5. The HCM$_{ROMS}$ reproduces the three-dimensional (3D) evolution of ENSO-related anomalies, revealing that the most pronounced temperature anomalies occur beneath the surface at 150 m. The interannual temperature anomaly budget highlights the dominance of the advection process in the simulated ENSO. Vertical mixing contributes negatively to ENSO anomalies, damping temperature anomalies from the surface due to the turbulent heat flux feedback. This newly developed HCM$_{ROMS}$ is poised to serve as an efficient modeling tool for ENSO research in the future.

## 1 Introduction

The El Niño and Southern Oscillation (ENSO), characterized by a warm phase (El Niño) and a cold phase (La Niña) sea surface temperature, occurs approximately every 2-7 years, representing the most prominent interannual climate variation mode on the Earth. ENSO greatly influences human and natural systems, reshaping global atmospheric circulation and weather systems (Yeh et al., 2018). It leads to extreme weather and climate events such as floods, droughts, and heat waves





(McPhaden et al., 2006). Thus, the accurate prediction of ENSO holds vital importance across various application sectors, including agricultural production, food security, freshwater resources, and the economy worldwide.

ENSO originates in the tropical Pacific through coupled ocean and atmosphere interactive processes (Bjerknes, 1969; Zebiak and Cane, 1987). The onset of ENSO is triggered by the reinforcing "Bjerknes feedback" among wind, sea surface temperature (SST), and the thermocline. During El Niño, an initial positive SST anomaly (SSTA) in the eastern Pacific reduces the east-west SST gradient, weakening the Walker circulation and thus the reduction of the easterly trade winds (Lindzen and Nigam, 1987). The weaker trade winds have two impacts on the ocean. On one hand, it reduces the equatorial upwelling in the eastern Pacific, causing the warm SST anomaly over the eastern basin. On the other hand, the weaker trade winds facilitate the eastward migration of surface warm water from the western Pacific warm pool. The eastward migration of surface warm water raises SST east of the dateline, triggering the eastward shift of deep convection. This, in turn, further relaxes the trade winds to the west of the convective center, causing positive feedback on the SST rise. During the cold phase of ENSO, La Niña's onset is the same as El Niño but with the opposite sign (Latif et al., 1994).

The phase transition of ENSO (i.e., the transition between El Niño and La Niña) is more complex. ENSO cycle can be affected by the wind-driven Kelvin waves, and the reflected Kelvin and Rossby waves, respectively, at the western and eastern boundaries of the Pacific, discharge-recharge processes due to Sverdrup transport, and anomalous zonal advection. These processes are named, in turn, in previous studies as the Western Pacific oscillator (Wang et al., 1999; Weisberg and Wang, 1997), the delayed oscillator (Battisti and Hirst, 1989; Suarez and Schopf, 1988), the recharge oscillator (Jin, 1997a, b), and the advective-reflective oscillator (Picaut et al., 1997). All these oscillators can work together as a unified oscillator to affect the ENSO cycle (Wang, 2001). Extratropical processes also affect ENSO as the thermal and salinity anomalies in the northern Pacific can be conveyed to the tropical Pacific along the subtropical cell (Gu and Philander, 1997; Kleeman et al., 1999; McCreary and Lu, 1994; Zhang et al., 1998; Zhou and Zhang, 2022a). In addition, ENSO can be affected by various forcing and feedback processes such as stochastic atmospheric forcing (Jin et al., 2007; Moore and Kleeman, 1999; Zhang et al., 2008), freshwater flux (Gao et al., 2020; Kang et al., 2014; Zhang et al., 2012), ocean biology-induced feedback (Shi et al., 2023; Tian et al., 2020; Zhang et al., 2018b), and tropical instability waves (Tian et al., 2019; Zhang, 2016; Zhang et al., 2023). Given the intricate interplays among these effects, comprehending the individual and collective impacts of these processes on ENSO poses a great challenge.

Numerical models are powerful tools for investigating and predicting ENSO. Over the past decades, a variety of coupled models, encompassing different levels of complexity in both ocean and atmosphere components, have been developed for the ENSO studies. These models can be categorized according to their approaches, such as the Statistical Model (SM), Harmonic Oscillator Model (HOM), Linear Inverse Model (LIM), Intermediate Coupled Model (ICM), Hybrid Coupled Model (HCM), Coupled General Circulation Model (CGCM), and Artificial Intelligence (AI); their characteristics are listed in Table 1. Each of these models boasts unique advantages in the realm of ENSO study. In particular, the HCM, which combines a comprehensive ocean general circulation model (OGCM) with a simplified atmospheric model, excels in incorporating complete ocean dynamic and thermodynamic processes without exhibiting notable climate drift issues (Zhang



et al., 2020). The advantage of HCM is attributed to the representation of atmospheric forcing in anomaly form, where interannual perturbations are considered in the model. This can isolate the interannual forcing and feedback effects in a clear way, better enhancing the understanding of the ENSO mechanism (McCreary and Anderson, 1991; Zhang et al., 2020).

In a previous study, Zhang (2015) developed an HCM (Z15 hereafter) based on an OGCM and a statistical atmospheric
model for interannual wind stress. The Z15 model successfully reproduces the quasi-four-year oscillation characteristics of ENSO (Gao et al., 2020; Zhang, 2015). However, despite various advantages of the Z15 model, such as a simple model architecture and high computational efficiency, the simulated El Niño (indicated by the positive SSTA) in the Z15 model is located near the dateline, suggesting a Central-Pacific instead of the typical Eastern-Pacific El Niño in the Z15 model (Gao et al., 2020; Zhang et al., 2018a). This is partly due to the use of the Gent-Cane OGCM (Gent and Cane, 1989) in the Z15
model, which is a reduced gravity ocean model with a limited ability to simulate deep temperature variations. The uppermost layer of the Gent-Cane OGCM is treated as a uniform mixed layer, with its depth determined by a bulk mixing parameterization (Chen et al., 1994). The bulk layer constraint hinders the Z15 model's ability to simulate vertical structure within the mixed layer. In addition, the Gent-Cane OGCM exhibits a restricted ability to simulate multi-scale dynamic processes, such as ocean mesoscale eddies and tropical instability waves. All the aforementioned underscore the urgent need
to replace the Gent-Cane OGCM in the Z15 model with an advanced ocean model to better simulate and investigate the ENSO mechanism. Therefore, in this study, we attempt to develop a new HCM based on a state-of-the-art Regional Ocean Modeling System (ROMS) model. Such an HCM$_{ROMS}$ can provide more detailed 3D ocean structure changes during ENSO evolution and help to explore the ENSO mechanism.

The objective of this study is to introduce the newly developed HCM$_{ROMS}$ and assess its performance. The paper is structured
as follows: Section 2 introduces the HCM$_{ROMS}$, including its statistical atmospheric model component, the ROMS setting, the HCM framework, and the numerical experimental design used in this study. Section 3 illustrates the HCM$_{ROMS}$ performance. Summaries are provided in Section 4.

**Table 1: List of the coupled models for ENSO Investigation.**

| Name | Method | Complexity* | Example |
|---|---|---|---|
| Statistical Models | Statistic relation between oceanic and atmospheric variables | 0 | Complex Empirical Orthogonal Function (CEOF) (Barnett, 1983, 1991)<br>Canonical Correlation Analysis (CCA) (Graham et al., 1987a, 1987b)<br>Principal Oscillation Patterns (POPs) (Hasselmann, 1988; von Storch et al., 1988) |
| Harmonic Oscillator Models | Ordinary differential equations | 1 | Delayed oscillator (Battisti & Hirst, 1989; Suarez & Schopf, 1988)<br>Recharge oscillator (F.-F. Jin, 1997a, 1997b)<br>Western Pacific oscillator (Wang et al., 1999; Weisberg & Wang, 1997)<br>Advective-reflective oscillator (Picaut et al., 1997)<br>Unified oscillator (Wang, 2001) |
| Linear Inverse Models | Stochastic differential equations | 1.5 | Linear inverse model (Newman et al., 2011; Penland & Sardeshmukh, 1995) |



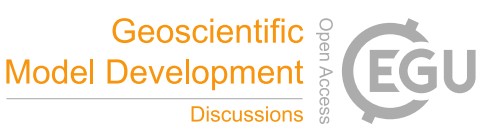

| Intermediate Coupled Models | Simplified ocean model coupled with statistical atmospheric model | 2 | ZC87 (Zebiak & Cane, 1987)<br>BH89 (Battisti & Hirst, 1989)<br>JN93 (F.-F. Jin & Neelin, 1993a, 1993b)<br>IOCAS-ICM (Zhang & Gao, 2016) |
|---|---|---|---|
| Hybrid Coupled Models | Complex ocean model coupled with statistical atmospheric model | 3 | N90 (J. D. Neelin et al., 1992; J. David Neelin, 1990)<br>B93 (Barnett et al., 1993)<br>S95 (Syu et al., 1995)<br>Z15 (Zhang, 2015; Zhang et al., 2018a) |
| Coupled General Circulation Models | Complex ocean model coupled with complex atmospheric model | 6 | CESM (National Center for Atmospheric Research, USA)<br>HadCM (Met Office Hadley Centre, UK)<br>MPI-ESM (Max Planck Institute for Meteorology, Germany)<br>BCCR-BCM (Bjerknes Centre for Climate Research, Norway)<br>FGOALS (LASG/IAP Chinese Academy of Sciences, China) |
| Artificial Intelligence Models | Artificial Intelligence | Infinity | Convolutional Neural Network (CNN) (Ham et al., 2019, 2021)<br>Residual CNN (Res-CNN) (J. Hu et al., 2021)<br>Air-sea coupler (ASC) based on the graph (Mu et al., 2021)<br>POP-Net (L. Zhou & Zhang, 2022)<br>3D-Geoformer (L. Zhou & Zhang, 2023) |

**\*The complexity is defined by the degree of freedom of the model variable in phase space, the oceanic and atmospheric complexity can be added together.**

## 2 Numerical Model and Experimental Design

### 2.1 Statistical Atmospheric Model

Following the Z15 modeling framework, a statistical atmospheric model is developed to represent the interannual perturbation fields of the atmosphere using the singular value decomposition (SVD) analysis (Zhang, 2015). Fig. 1 illustrates the schematic diagram of the statistical atmospheric model. The coupling relationship between any two interannual variation signals, such as SSTA and wind stress anomaly ($\mathbf{X}$ and $\mathbf{Y}$ in Fig. 1a), can be established by decomposing their covariance matrix using SVD (Fig. 1a). In an SVD analysis, the left-singular vector ($\mathbf{L}$ in Fig. 1a) and right-singular vector ($\mathbf{R}$ in Fig. 1a)

of the covariance matrix ($\mathbf{S}$ in Fig. 1a) serve as eigenvectors for the left field $\mathbf{X}$ and right field $\mathbf{Y}$, respectively (Fig. 1a). Notably, the left-singular vector $\mathbf{L}$ and right-singular vector $\mathbf{R}$ share the same singular value $\sum$. Therefore, given a specific left field ($\mathbf{Xm}$), the corresponding right field signal ($\mathbf{Ym}$) can be determined by SVD relation through the inversion process shown in Fig. 1b.



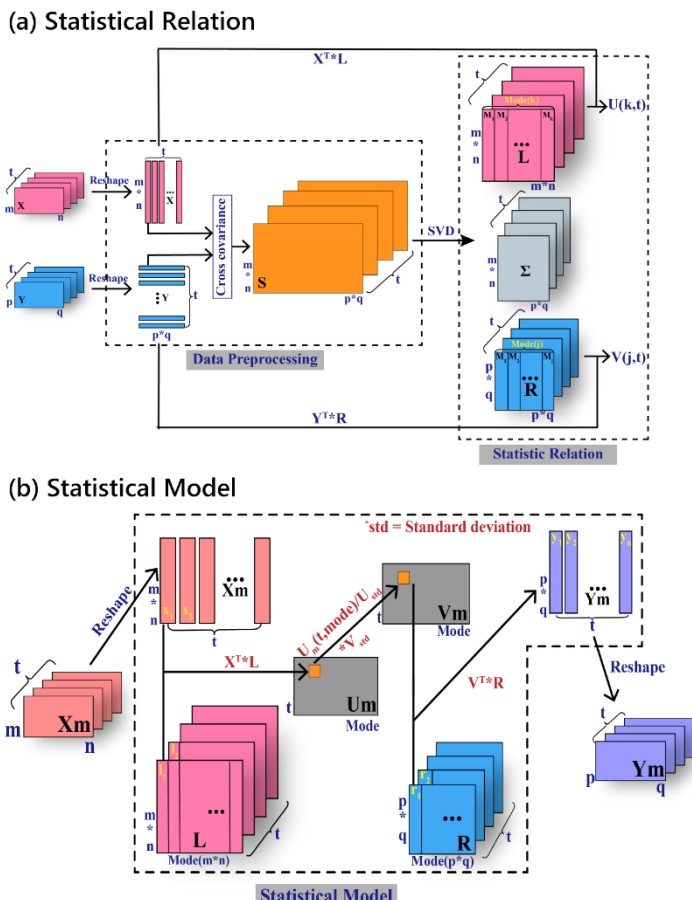

**Figure 1: Flow chart of (a) SVD analysis and (b) SVD-based statistical model. X and Y in (a) are original interannual variation signals before SVD, S is the covariance matrix between X and Y, L is the left-singular vector and R is the right-singular vector, U and V are, respectively, the time series of left-singular vector and right-singular vector, and $\sum$ is the singular value matrix. Xm in (b) is the input left field to the statistical model, Ym is the output right field, and Um and Vm are calculated time series of left field and left field in the statistical model, respectively.**

To include the ENSO-related atmospheric forcing-feedback processes in HCM$_{ROMS}$, we established statistical relations between the interannual SSTA ($SST_{inter}$) and both interannual wind stress anomalies ($\tau_{inter}$) and interannual freshwater flux anomalies ($FWF_{inter}$) using the SVD analysis (Fig. 1a). It should be noted that the freshwater flux (FWF) in this work is defined as the net freshwater flux into the ocean, which is represented by the total precipitation (P) minus evaporation (E), i.e., P-E. The monthly $SST_{inter}$ used to construct empirical statistical relations are derived from the National Oceanic and Atmospheric Administration (NOAA) Optimum Interpolation (OI) SST dataset with a spatial resolution of 1°×1° from 1981 to 1999 (Reynolds et al., 2002). The monthly $\tau_{inter}$ and $FWF_{inter}$ come from a 24-member ensemble of the European Centre Hamburg Model (ECHAM) version 4.5 simulations forced by observed SST from 1950 to 1999 (Roeckner et al.,





1996). The use of the ensemble-averaged $\tau_{inter}$ and $FWF_{inter}$ from the ECHAM simulations is to reduce the impacts of

atmospheric noise (Zhang et al., 2003).

The first SVD modes for $SST_{inter}$, $\tau_{inter}$, and $FWF_{inter}$ are shown in Figs. 2a, 2b, and 2c, respectively. It shows that the

SVD patterns include ENSO-related forcing and response processes. For example, the El Niño-type SSTA in the equatorial

eastern Pacific (Fig. 2a) is associated with the tropical westerly wind stress anomalies (Fig. 2b) and FWF anomalies around

the dateline (Fig. 2c). We calculated the squared covariance fractions of different SVD modes. The first five SVD modes can

contribute 99.4% of the $\tau_{inter}$ variations and 97.6% of the $FWF_{inter}$ variations. Therefore, we only include the first five

SVD modes in the statistical atmospheric model to reproduce the interannual atmospheric forcing. The performance of the

statistical atmospheric model in reproducing the $\tau_{inter}$ and $FWF_{inter}$ are evaluated in section 3.1.

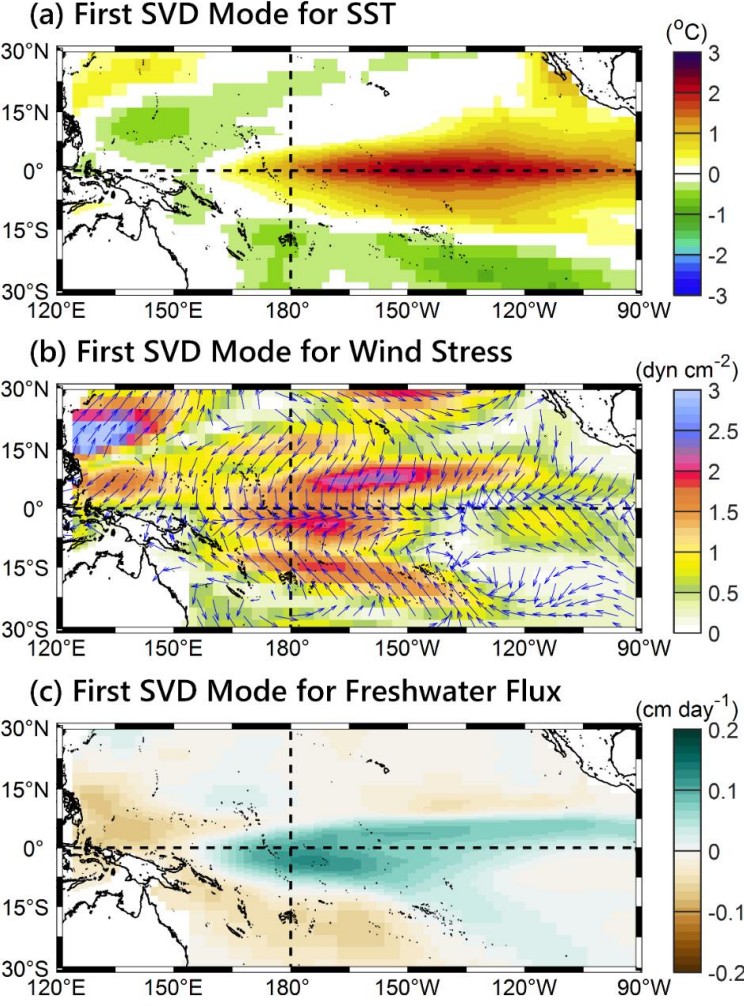

**Figure 2: First SVD modes for (a) interannual SSTA, (b) interannual wind stress anomaly, and (c) interannual freshwater flux**
**anomaly.**



## 2.2 The Regional Ocean Modeling System (ROMS) Model

The comprehensive ocean model used in HCM$_{ROMS}$ is based on the Regional Ocean Modeling System (ROMS) model (Shchepetkin and Mcwilliams, 2005). ROMS is a community regional ocean model suitable for various applications in ocean forecasting and research needs. ROMS incorporates accurate and efficient physical and numerical algorithms. For example, all 2D and 3D equations in ROMS undergo time discretization using a third-order predictor and corrector time-stepping algorithm (Shchepetkin and Mcwilliams, 2005). The enhanced stability of the scheme allows larger time steps, which is important for climate simulations that need to be integrated for long periods. In addition, ROMS employs an explicit time-splitting scheme to solve the hydrostatic primitive equations for momentum. Within each baroclinic (3D; solving the primitive equations) time step, a finite number of barotropic (2D; solving the shallow water equations) time steps are executed to evolve the free-surface and vertically integrated momentum equations. The time-splitting method allows the ROMS model to simulate and ensure the stability of high-frequency waves, increasing the model's accuracy in simulating the sea level variations.

Fig. 3 shows the domain setting of the ROMS model used in this study. The ROMS model with a horizontal resolution of 0.5°×0.5° (~52.9 km × 52.9 km) covers the tropical Pacific ranging from 95 °E to 70 °W and 30 °S to 30 °N. The ROMS model has 50 layers in the vertical direction, with a higher resolution (~0.1 m on the model grid with a water depth of 5000 m) near the sea surface using a terrain-following "S" coordinate with Shchepetkin's double stretching function (Shchepetkin and Mcwilliams, 2009). The ROMS parametrizations include the nonlocal K-profile scheme for vertical mixing (Large et al., 1994) and the Smagorinsky scheme for horizontal diffusion (Smagorinsky, 1963). The short-wave radiation penetration into the ocean is calculated by a double exponential irradiance absorption scheme (Paulson and Simpson, 1977), with Jerlov water type I parameters over the open ocean and Jerlov water type II parameters over the marginal seas (Kuo et al., 2023; Yu et al., 2017, 2020, 2022).

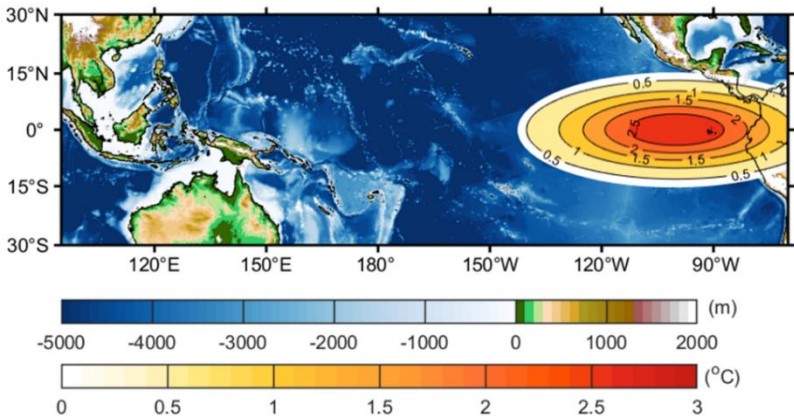

**Figure 3: ROMS model domain and topography. The contours with orange shading show the El Niño-like SSTA which drives the statistical atmospheric model during the "initial kick for eight months".**



## 2.3 Hybrid Coupled Model Framework

Fig. 4 illustrates the schematic diagram of the HCM$_{ROMS}$. It consists of the statistical atmospheric model and the ROMS model. The ROMS model is driven by the wind stress ($\tau$) and FWF at the air-sea interface. The $\tau$ and FWF in the HCM$_{ROMS}$ can be written as $\tau = \tau_{clim} + \alpha_\tau \tau_{inter}$ and $FWF = FWF_{clim} + \alpha_{FWF} FWF_{inter}$, respectively, where $\tau_{clim}$ and $FWF_{clim}$ are prescribed climatological forcing, $\tau_{inter}$ and $FWF_{inter}$ are interannual perturbations, and $\alpha_\tau$ and $\alpha_{FWF}$ are coupling coefficients of the interannual perturbations. The coupling coefficients $\alpha_\tau$ and $\alpha_{FWF}$ are designed to counteract the reduction in retrieved $\tau_{inter}$ and $FWF_{inter}$ caused by the linear constraints within the statistical atmospheric model. The application of coupling coefficients also provides a method to investigate the sensitivity of the ENSO evolution to interannual perturbations by using different $\alpha_\tau$ and $\alpha_{FWF}$ values. The climatological forcing $\tau_{clim}$ and $FWF_{clim}$ are derived from the climatological atmospheric reanalysis data. While the interannual perturbations $\tau_{inter}$ and $FWF_{inter}$ are calculated using the statistical atmospheric model forced by the ROMS-simulated $SST_{inter}$. The $SST_{inter}$ in the ROMS model can be obtained by subtracting the climatological SST ($SST_{clim}$) from the ROMS-simulated SST. It should be noted that the $SST_{clim}$ here is derived from the ROMS-simulated SST forced by the climatological forcing (e.g., $\tau_{clim}$ and $FWF_{clim}$), rather than being sourced from other origins. This ensures the model does not suffer from climate drift issues.

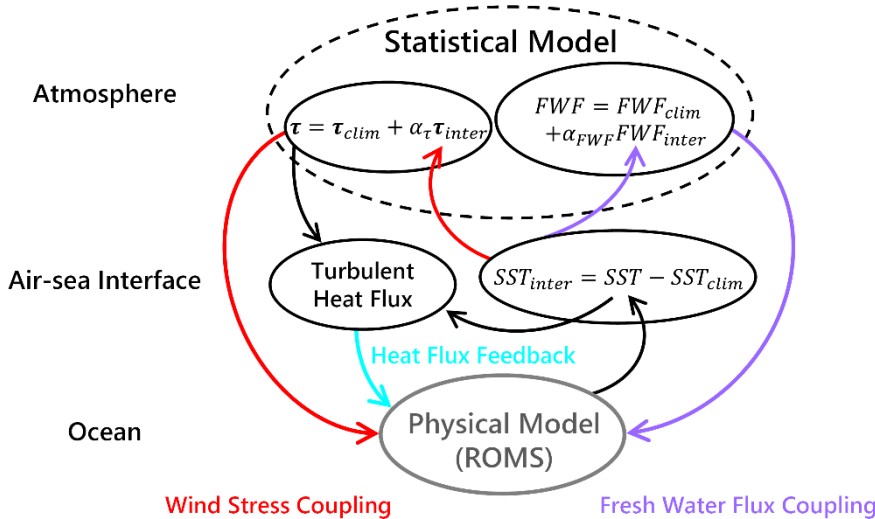

**Figure 4: Schematic diagram of the HCM$_{ROMS}$. Red arrows represent wind stress coupling, purple arrows represent the FWF coupling, and the blue arrow represents the heat flux feedback.**

It is also important to highlight the coupler in the HCM$_{ROMS}$ for the coupling process between the statistical atmospheric model and the ROMS model. Incorporating the statistical atmospheric model requires $SST_{inter}$ from the entire ROMS domain. Therefore, data transmissions between different computing nodes are necessary for the coupling process. A coupler module based on the Message Passing Interface (MPI) has been developed to facilitate the data transmissions in the





HCM$_{ROMS}$ (Fig. 5). With the coupler, the HCM$_{ROMS}$ gathers the $SST_{inter}$ from different computing nodes to the main computing node before coupling (red curves in Fig. 5). The main computing node then computes $\tau_{inter}$ and $FWF_{inter}$ for the

entire ROMS domain and sends $\tau_{inter}$ and $FWF_{inter}$ tiles to the corresponding computing nodes to drive the ROMS model (blue curves in Fig. 5). We have evaluated the performance of the coupler module to transfer data accurately and efficiently between the computing nodes (figs. not shown). The successful construction of this coupler module lays the groundwork for the HCM$_{ROMS}$ development.

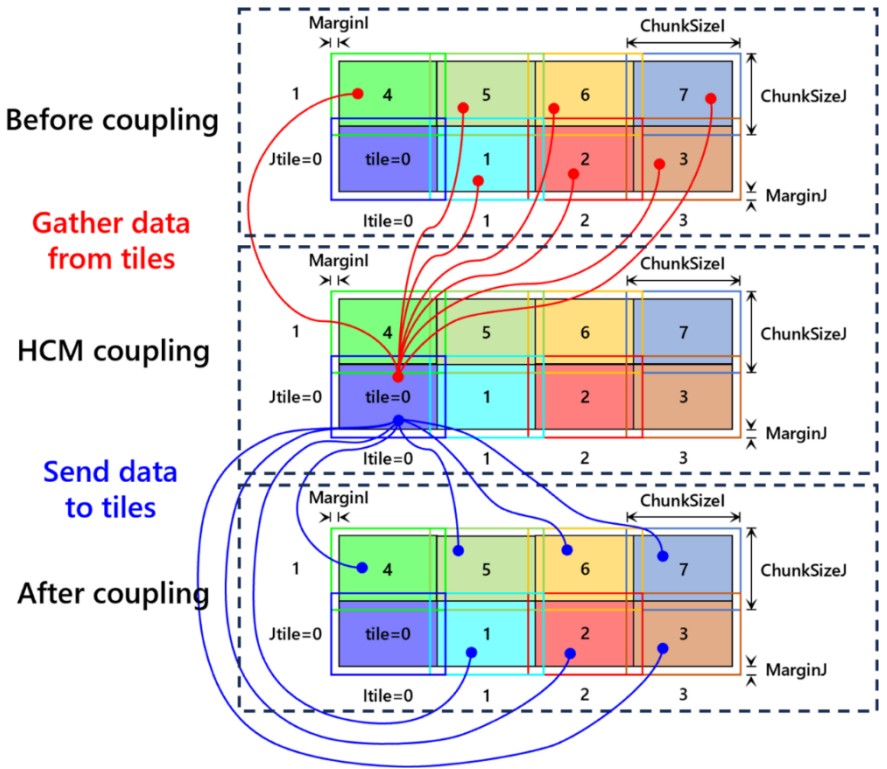

**Figure 5: Flow chart of the coupling process using MPI.**

Besides the statistical atmospheric model, we also introduce a module in the HCM$_{ROMS}$ to address the damping effects induced by the ocean-atmosphere heat transfer processes. The module based on a bulk approximate utilizes surface wind

speed derived from the wind stress inversion to compute the turbulent heat transfer between the ocean and atmosphere, including both latent and sensible heat fluxes (Fairall et al., 1996):

$$LH = \rho_a L_e C_E V_{wg}\big(q_a - 0.98 \times q_{sat}(SST)\big) \quad ---- (1)$$
$$SH = \rho_a c_p C_H V_{wg}(T_a - SST) \qquad\qquad ---- (2)$$




where $LH$ and $SH$ are latent and sensible heat fluxes, respectively; $\rho_a$ is dry air density; $L_e$ is the latent heat of vaporization;
$c_p$ is the specific heat of dry air at constant pressure; $C_E$ and $C_H$ are respectively the transfer coefficients for latent and
sensible heats and set to 1.4×10⁻³ in this study; $V_{wg}$ is the surface wind speed derived from the wind stress inversion ($V_{wg} = \sqrt{\frac{|\tau|}{\rho_a C_D}}$), where $|\tau|$ is the magnitude of the wind stress vector and $C_D$ is the transfer coefficient for drag with a value of
1.7×10⁻³; $T_a$ is air temperature; $q_a$ is the atmospheric specific humidity; and $q_{sat}(SST)$ is the saturated specific humidity at
SST.

## 2.4 Experimental Design

The HCM$_{ROMS}$ model was first integrated for 60 years using the climatological atmospheric forcing (i.e., $\alpha_\tau$=0 and $\alpha_{FWF}$=0)
as the model's spin-up. The spin-up run was initialized with the January climatology of the Simple Ocean Data Assimilation
Version 3 (SODA3) reanalysis with a horizontal resolution of 0.5°×0.5° (Carton et al., 2018). The climatological monthly
SODA3 data also served as the lateral boundary conditions of sea surface height (SSH), currents, temperature, and salinity
throughout the model integration. In terms of atmospheric forcing, the climatological monthly data for $\tau_{clim}$, $FWF_{clim}$, short
wave and long wave radiations were obtained from the Common Ocean Reference Experiment version 2 (COREv2) global
air-sea flux dataset with a horizontal resolution of 1°×1° (Large and Yeager, 2009). The climatological monthly data for sea
level pressure (SLP), surface air temperature, and surface air relative humidity, utilized in calculating surface latent and
sensible heat fluxes, were sourced from the International Comprehensive Ocean-Atmosphere Data Set (ICOADS) with a
horizontal resolution of 1°×1° (Freeman et al., 2017). It should be noted that the model's climatology in the following
sensitivity experiments is calculated over the last 10 years of the spin-up run.

After the spin-up, we executed five sensitivity experiments, adjusting the $\alpha_\tau$ values ($\alpha_\tau$=1.0, 1.3, 1.5, 1.7, and 2.0), to
determine the optimal $\alpha_\tau$ for reproducing sustainable interannual variabilities in the HCM$_{ROMS}$. These sensitivity
experiments were integrated for 30 years and initiated through a so-called "initial kick", where the HCM$_{ROMS}$ underwent
forced integration with an imposed westerly wind anomaly lasting eight months. The westerly wind anomaly during the
"initial kick" was created using the statistical atmospheric model driven by an El Niño-like SSTA (Fig. 3), which was
detected by the statistical atmospheric model but did not manifest in the simulated SST. Subsequent anomalous conditions
evolved solely through coupled ocean-atmosphere interactions within the HCM$_{ROMS}$. The FWF effects are not taken into
account in this study by setting $\alpha_{FWF}$=0. This is because of the primary impact of the SSTA-$\tau_{inter}$ coupling on shaping the
interannual variability associated with ENSO, while the SSTA-$FWF_{inter}$ coupling, although capable of affecting the ENSO
intensity, plays a secondary role (Gao et al., 2020). Further analysis of the FWF effects on the ENSO intensity in HCM$_{ROMS}$
will be presented in Part II of this study.





## 3. Results and Discussion

### 3.1 Statistical Atmospheric Model Performance

We first examine the performance of the statistical atmospheric model in reproducing the ENSO-related interannual atmospheric forcing. Fig. 6a shows the 40-year observed SSTA over the tropical zone of 5 °S to 5 °N from 1980 to 2020. The observed SSTA are derived from the monthly NOAA Extended Reconstructed SST (ERSST) version 5 dataset (Huang et al., 2017). We adopt ERSST in this subsection instead of using the NOAA OI SST, which is used to construct the statistical atmospheric model in section 2.1, to maintain the independence of the assessment. The observed SSTA shows a

clear interannual characteristic related to ENSO, especially during the three major El Niño events of 1982/1983, 1997/1998, and 2015/2016, where the positive SSTA over the eastern Pacific can be above 2 °C (Fig. 6a). The 40-year zonal wind stress anomaly and FWF anomaly over the tropical zone of 5 °S to 5 °N from 1980 to 2020 are shown in Figs. 6b and 6c, respectively. The monthly zonal wind stress anomaly and FWF anomaly are derived from the National Centers for Environmental Prediction and the National Center for Atmospheric Research (NCEP/NCAR) Reanalysis (Kalnay et al.,

1996). During El Niño, the positive SSTA in the eastern Pacific corresponds to the eastward expansion of the anomalous westerly winds originating from the dateline. While in La Niña, abnormal easterly winds near the dateline emerge, leading to a cold SSTA east of the dateline (Figs. 6a and 6b). As for the FWF, the positive SSTA in the eastern Pacific during El Niño shifts the precipitation eastward to increase FWF in the eastern equatorial Pacific and reduces FWF in the western Pacific. Conversely, during La Niña, the FWF in the western equatorial Pacific increases, while FWF in the eastern Pacific decreases

(Figs. 6a and 6c).

    The retrieved zonal wind stress anomaly and FWF anomaly using the statistical atmospheric model forced by the observed SSTA from 1980 to 2020 are shown in Figs. 6d and 6e, respectively. During the three major El Niño events of 1982/1983, 1997/1998, and 2015/2016, the model replicates the observed anomalous westerly winds originating from the dateline (Figs. 6b and 6d). Additionally, in the five major La Niña events of 1988/1989, 1998/1999, 1999/2000, 2007/2008, and 2010/2011,

the statistical atmospheric model also reproduces the abnormal easterly winds near the dateline, consistent with NCEP/NCAR reanalysis (Figs. 6b and 6d). As for the FWF, the statistical atmospheric model reproduces the ENSO-related FWF anomaly dipole. The statistical atmospheric model produces a higher (lower) FWF in the eastern (western) equatorial Pacific during El Niño but in the opposite during La Niña (Figs. 6c and 6e). Although the statistical atmospheric model tends to underestimate the strength of interannual perturbations, especially for the wind stress, which may be due to the linear

constraints in the SVD when dealing with nonlinear signals (Fig. 6b versus 6d). The statistical atmospheric model reproduces the observed interannual wind stress and FWF anomalies in phase (Figs. 6b-e). The correlation coefficient between the observed and simulated zonal wind stress anomalies is 0.62 (p<0.01). While the correlation coefficient for the observed and simulated FWF anomalies is 0.68 (p<0.01).



**Figure 6: Hovmöller diagram of observed (a) SSTA, (b) zonal wind stress anomaly, and (c) FWF anomaly over the tropical zone of 5 °S to 5 °N from 1980 to 2020. (d) and (e) are respectively the retrieved zonal wind stress anomaly and FWF anomaly using the statistical atmospheric model forced by the observed SSTA. The vertical dashed line indicates the dateline.**



## 3.2 Simulated Climatology in the Ocean

The ROMS model performance in simulating the long-term mean and seasonal variability of ocean temperatures in the tropical Pacific were also assessed by comparing the simulated ocean climatology with observational data. Here we note again that the simulated ocean climatology in the study is defined as the long-term monthly mean averaged over the last 10 years of the spin-up. This prolonged duration is necessary as the ROMS model takes 30 to 40 years for the simulated heat content in the upper 2000 m to reach equilibrium (figs. not shown). To mitigate potential climate drift concerns arising from the model not reaching equilibrium, we employed a dataset covering the 51 to 60 years of the spin-up run to calculate the model's climatology. The observed climatological monthly ocean temperature comprises averaged values from the World Ocean Atlas (WOA) 2023 dataset over the observation period of 1955 to 2022, with a horizontal resolution of 1°×1° (Locarnini et al., 2023).

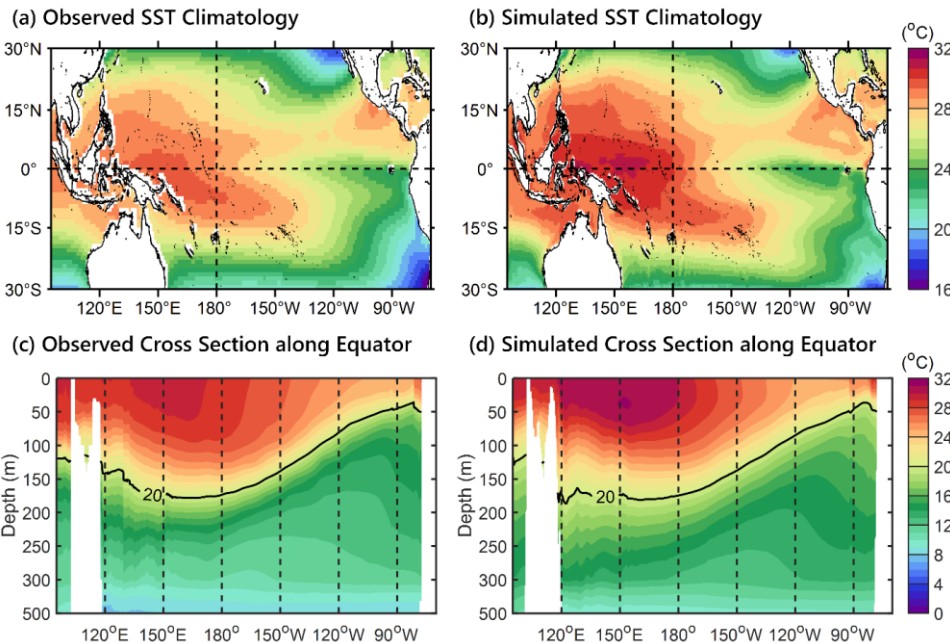

**Figure 7: Horizontal distribution of (a) observed and (b) simulated long-term mean SST over the tropical Pacific. The (c) observed and (d) simulated vertical cross-section of the temperature along the equator.**

Fig. 7a shows the horizontal distribution of observed long-term mean SST over the tropical Pacific. It shows that the observed SST in the western Pacific warm pool with an average value of 30 °C is notably higher than in the surrounding area. Conversely, the cold tongue region of the eastern equatorial Pacific shows distinct upwelling characteristics, leading to a lower average SST of only 24 °C (Fig. 7a). The observed SST distribution matches the subsurface temperature structure changes along the equator. The vertical cross-section of the observed temperature over the tropical zone of 5 °S to 5 °N





along the equator is shown in Fig. 7c. The higher SST observed in the western Pacific warm pool is attributed to the westward water transport in the equatorial Pacific, influenced by trade winds. The accumulated surface warm water in the western Pacific warm pool descends, leading to a deepening of the 20 °C isotherm, reaching its maximum depth of 180 m west of the dateline. In the tropical eastern Pacific, the upwelling compensates for the trade-wind-induced westward surface transport, resulting in the ascent of deep cold water to the surface. The depth of the 20 °C isotherm progressively decreases

away from the dateline, reaching its minimum depth of only 50 m along the coast of Peru at 90 °W (Fig. 7c).

The simulated long-term mean SST distribution and vertical temperature structure along the equator are shown in Figs. 7b and 7d, respectively. The ROMS model replicates the observed long-term mean temperature in the tropical Pacific as indicated by the WOA data. The simulated western Pacific warm pool and the eastern Pacific cold tongue closely resemble those observed patterns (Figs. 7a and 7b). However, there is a notable overestimation of approximately 1 °C in the SST of

the western Pacific warm pool by the ROMS model (Fig. 7b). This discrepancy may be attributed to the climatological atmospheric forcing adopted in the model, which excludes the influence of high-frequency stochastic atmospheric forcing signals such as tropical cyclones, westerly wind bursts, and Madden-Julian oscillations. The weaker climatological forcing, particularly from surface winds, leads to less vigorous vertical mixing in the ocean, potentially resulting in a warm bias in simulated SST. It is important to note that the modeled climatology, driven by climatological forcing, still exhibits some

differences from the observed climatology, which refers to climatological results. The ROMS model reproduces the observed mean vertical temperature structure along the equator (Figs. 7c and 7d). Same as the WOA observation, the simulated 20 °C isotherm deepens in the western equatorial Pacific, with a maximum depth of 180 m west of the dateline. As moving away eastward from the dateline to the eastern equatorial Pacific, the simulated 20 °C isotherm depth decreases, reaching a minimum depth of 50 m along the coast of Peru at 90 °W (Figs. 7c and 7d).

Fig. 8 illustrates the seasonal climatology of observed and simulated SST in the tropical zone of 5 °S to 5 °N. West of the dateline, the observed SST exhibits a semi-annual pattern. The highest temperature, reaching 30.5 °C in both May and November, while the lowest temperature 29.5 °C, appears in February and August (Fig. 8a). This semi-annual SST variation in the western equatorial Pacific is attributed to the sun's biannual crossing of the equator, leading to corresponding semi-annual changes in solar radiation. However, the observed SST shows a distinct annual cycle in the eastern Pacific, with the

highest temperature of 28 °C in March and the lowest temperature of 23 °C in September (Fig. 8a). The annual SST variation in the eastern equatorial Pacific is driven by the effects of trade-wind-induced upwelling. The annual strengthening and weakening of the trade winds contribute to the annual SST changes in the eastern equatorial Pacific. The simulated seasonal SST climatology is shown in Fig. 8c. Despite the ROMS model overestimating the average SST in the western Pacific warm pool by 1 °C (Figs. 7b and 8c), it captures the observed semi-annual SST variation in the western equatorial Pacific and the

annual SST variation in the eastern equatorial Pacific. The annual variations in both observed and simulated SST are shown in Figs. 8b and 8d, respectively. The ROMS model replicates the observed seasonal SST changes, demonstrating an annual SST variation amplitude of ±3 °C in the eastern equatorial Pacific and a semi-annual SST variation amplitude of ±0.5 °C in the western equatorial Pacific.



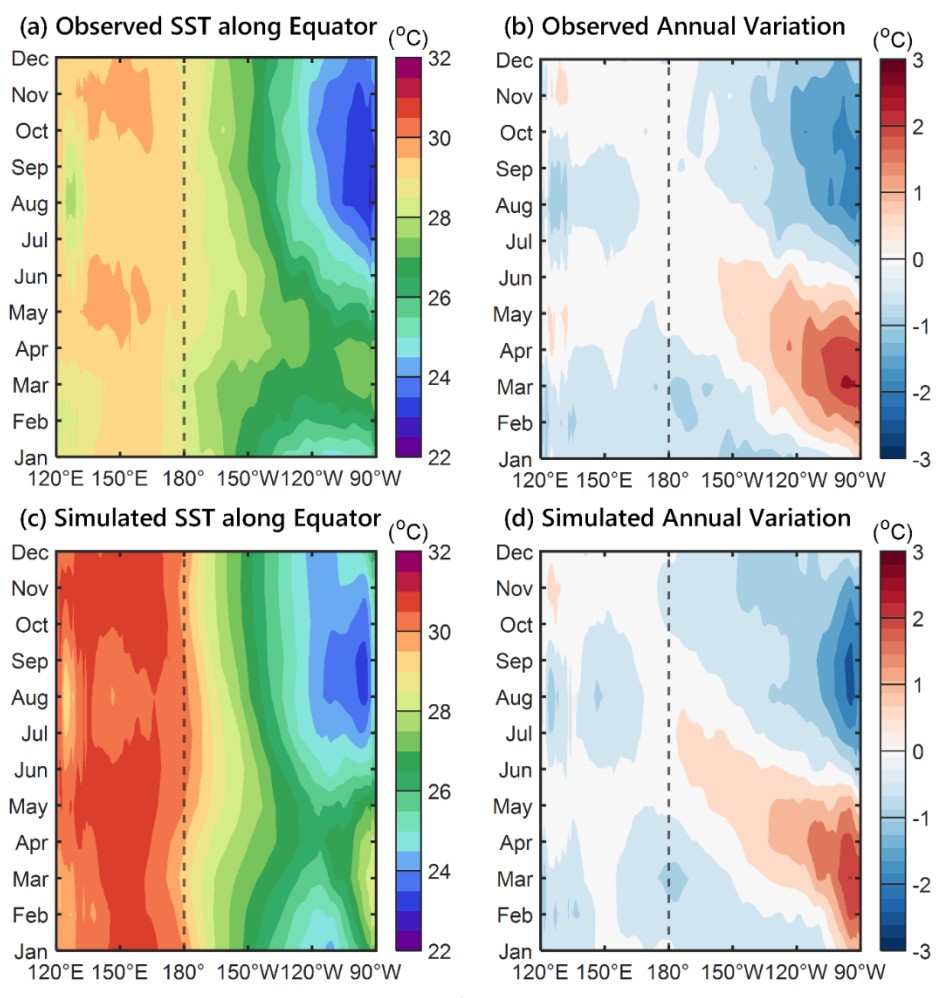

**Figure 8: Hovmöller diagram of (a) observed and (c) simulated seasonal SST along the equator. (b) and (d) are their annual variations.**

### 3.3 Simulated Interannual Variability associated with ENSO

Fig. 9 shows the simulated Niño 3.4 index (SSTA averaged over the box region of 5 °S to 5 °N, 170 °W to 120 °W) in the sensitivity experiments with different interannual wind stress coupling coefficients ($\alpha_\tau$=1.0, 1.3, 1.5, 1.7, and 2.0). Changes in the coupling coefficient $\alpha_\tau$ lead to alterations in the simulated interannual SST variability. The statistical atmospheric model tends to underestimate the strength of interannual disturbances due to the linear constraints within SVD (Fig. 6). Therefore, if the linear constraint effects are not counteracted, i.e., $\alpha_\tau$=1.0, the initial interannual oscillation signals generated during the "initial kick" will rapidly dissipate over 36 months (purple line in Fig. 9). This dissipation effect



diminishes gradually as the coupling coefficient $\alpha_\tau$ increases. When $\alpha_\tau$=1.3, the initial interannual oscillation signals gradually weaken and dissipate over 120 months (blue line in Fig. 9), whereas $\alpha_\tau$=1.5, the initial interannual oscillation signals can be consistently sustained, exhibiting a steady quasi-three-year cycle (black thick line in Fig. 9). However, when $\alpha_\tau$ exceeds 1.5, with increases in the coupling coefficient $\alpha_\tau$, the simulated interannual oscillations become unstable. At

$\alpha_\tau$=1.7, the amplitude of the simulated Niño 3.4 index increases to 2.8 °C, exceeding the initial amplitude of 1.8 °C (red line in Fig. 9). Meanwhile, at $\alpha_\tau$=2.0, the period of the simulated interannual SST variability changes, the stable quasi-three-year oscillations give way to irregular quasi-biennial oscillations, which are characterized by alternating strong and weak biennial oscillations over a four-year period (orange line in Fig. 9). The above evidence suggests a substantial relation between the modeled interannual SST and the coupling coefficient $\alpha_\tau$. Given that the optimal $\alpha_\tau$=1.5 produces sustainable interannual

variabilities in the HCM$_{ROMS}$, subsequent analysis is based on the experiment with the interannual wind stress coupling coefficient $\alpha_\tau$ set at 1.5.

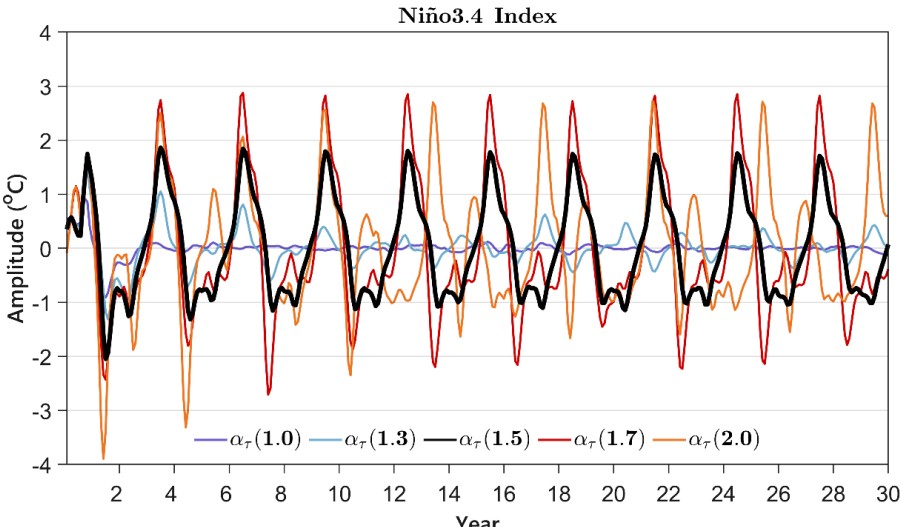

**Figure 9: Simulated Niño 3.4 index in the sensitivity experiments with different interannual wind stress coupling coefficients. The**
**purple line is for $\alpha_\tau$=1.0, the blue line for $\alpha_\tau$=1.3, the black line for $\alpha_\tau$=1.5, the red line for $\alpha_\tau$=1.7, and the orange line for $\alpha_\tau$=2.0, respectively.**

The Hovmöller diagrams of simulated SSTA, zonal wind stress anomaly, surface height anomaly, and surface heat flux anomaly for the sensitivity experiment with $\alpha_\tau$=1.5 are shown in Figs. 10a, 10b, 10c, and 10d, respectively. These anomalies
are averaged over the tropical zone of 5 °S to 5 °N along the equator. During the first-eight-month "initial kick", the prescribed idealized El Niño-like SSTA (Fig. 3) drives the statistical atmospheric model to generate abnormal westerly winds east of the dateline (Fig. 10b). The anomalous westerly winds persist for eight months, transporting surface warm water eastward and increasing the sea surface height in the eastern equatorial Pacific (Fig. 10c). The eastward water transport



induced by anomalous westerly winds during the "initial kick" drives the ROMS model to produce an initial positive SSTA

of 2 °C east of the dateline (Fig. 10a). The initial positive SSTA in the eastern Pacific subsequently triggers the ENSO cycle in the HCM$_{ROMS}$. Specifically, the simulated ENSO in the HCM$_{ROMS}$ shows alternating occurrence of El Niño with a positive SSTA of 2 °C and La Niña with a negative SSTA of -1 °C, maintaining a stable quasi-three-year cycle (Fig. 10a). During the simulated El Niño and La Niña, anomalous westerly wind stress anomaly of 0.3 dyn cm$^{-2}$ and easterly wind stress anomaly of -0.3 dyn cm$^{-2}$ emerge around the dateline, the same as the observations (Figs. 6b and 10b). The abnormal westerly and

easterly winds that occur near the dateline during El Niño and La Niña strengthen the eastward and westward equatorial water transport, contributing to the SSTA evolution associated with El Niño and La Niña, respectively (Fig. 10c). In addition, the surface heat flux anomaly with an amplitude of ±60 W/m$^2$ is seen to dampen the positive and negative SSTAs associated with the simulated El Niño and La Niña, respectively (Fig. 10d).

We note that incorporating the SSTA-$\tau_{inter}$ coupling in the HCM$_{ROMS}$ can reduce the average SST in the western Pacific

warm pool by 0.6~0.8 °C (Fig. 10a). This helps alleviate the overestimation of SST during the model spin-up due to the weaker climatological atmospheric forcing adopted in the model (Figs. 7 and 8), implying that ENSO may play a certain role in regulating the average state of the western Pacific warm pool and Pacific western boundary currents (Hu et al., 2015). Although it is not the focus of this study, the newly developed HCM$_{ROMS}$ may serve as a potential tool to investigate the ENSO impacts on the western Pacific warm pool and Pacific western boundary currents in the future.


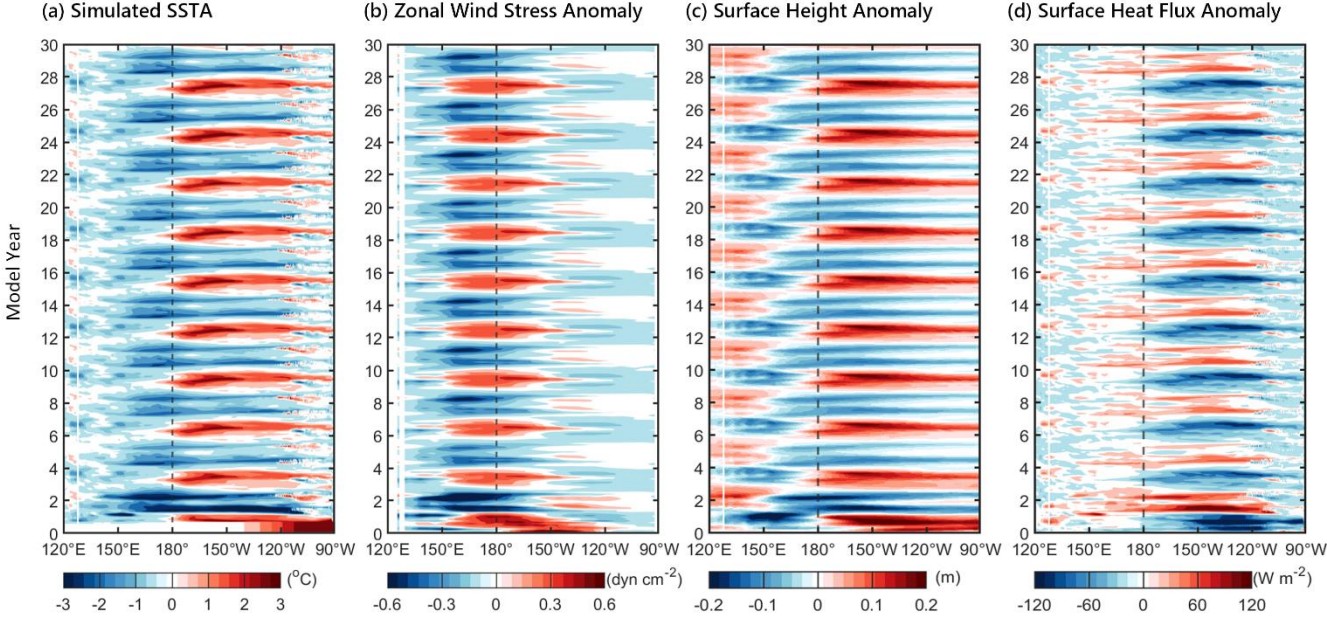

**Figure 10: Hovmöller diagram of simulated (a) SSTA, (b) zonal wind stress anomaly, (c) surface height anomaly and (d) surface heat flux anomaly averaged over the tropical zone of 5 °S to 5 °N along the equator for the sensitivity experiment with $\alpha_\tau$=1.5.**



To assess the simulated ENSO in the HCM$_{ROMS}$, following the method documented by Timmermann et al. (2018), we conducted an Empirical Orthogonal Function (EOF) analysis on the simulated SSTA from 9 to 30 years (three cycles after the "initial kick") of the sensitivity experiment with $\alpha_\tau$=1.5. It shows that the leading EOF (Mode 1), contributing 49.97% to the variance, has a classic El Niño pattern (Fig. 11a) and exhibits variability on a quasi-three-year timescale (blue line in Fig. 11c). The second EOF (Mode 2), contributing 14.31% to the variance, presents a tropical east-west zonal dipole (Fig. 11b)

with enhanced variability on a 1.5-year timescale (orange line in Fig. 11c). The interplay between the first and second EOFs captures the spatial and temporal evolution of the simulated ENSO (Figs. 11d-g). The second EOF serves as an amplifier for the simulated warming anomaly in the equatorial eastern Pacific during El Niño, emphasizing the impacts of the Bjerknes feedback. Specifically, when the simulated El Niño onset (time coefficient of Mode 1 surpassing 1), the time coefficient of Mode 2 undergoes a transition from positive to negative, reaching its minimum of -2 one month after the time coefficient of

Mode 1 achieving its maximum of 1.5 (Fig. 11c). The opposite-sign pattern of Mode 2 corresponds to the process wherein the Bjerknes feedback induces warming in the eastern equatorial Pacific and cooling in the western equatorial Pacific during El Niño, implying the second EOF may represent the effects of the Bjerknes feedback (Fig. 11b). However, this amplifying effect of the second EOF was absent during the simulated La Niña. Both Mode 1 and Mode 2 shared the same negative sign during the onset of simulated La Niña (time coefficient of Mode 1 falling below -1; Fig. 11c), with Mode 2 acting as a

damper rather than amplifying the cooling in the eastern equatorial Pacific (Fig. 11f). This explains the asymmetry between simulated El Niño and La Niña in the HCM$_{ROMS}$, where the amplitude of the simulated Niño 3.4 index during El Niño surpasses that during La Niña (Figs. 9 and 10a).

As a comparison, the first and second EOFs of the observed SSTA from 1980 to 2020 are shown in Figs. 12a and 12b, respectively. Notably, the simulated second EOF in HCM$_{ROMS}$ fails to capture the influence of the Victoria mode compared

to the observation (Figs. 11b and 12b). This discrepancy may arise from the limited model domain of the tropical Pacific (30 °S to 30 °N; Fig. 3) used in this study, which results in the absence of extratropical processes in the HCM$_{ROMS}$ (Ding et al., 2017, 2019). Nevertheless, the EOFs of observed and simulated SSTA still share certain similarities. The first EOF of observed SSTA shows an El Niño pattern with a variance contribution of 43.90% (Fig. 12a) and the second EOF presents a tropical east-west zonal dipole with a variance contribution of 11.11% (Fig. 12b). In addition, the observed first and second

EOFs have similar configurations (phase configuration of Mode 1 and Mode 2; the phase vectors shown in Figs. 12c-f) during El Niño and La Niña compared to the simulated EOFs (Figs. 11d and 11f). During the 1997/1998 and 2015/2016 El Niño events, the observed SSTAs feature positive Mode 1 and negative Mode 2 (phase vectors are in the third quadrant; Figs. 12c and 12e), while during the 1999/2000 and 2010/2011 La Niña events, they exhibit both negative Mode 1 and Mode 2 (phase vectors are in the fourth quadrant; Figs. 12d and 12f), highlighting the asymmetrical role of Mode 2 during El Niño

and La Niña. While the HCM$_{ROMS}$ depicts the distinct functions of the second EOF during the simulated El Niño and La Niña (Figs. 11c, 11d, and 11f), complex factors may contribute to the mechanisms involved, which surpass the scope of this study. We currently abstain from further investigation into details, merely speculating that the asymmetry in the Bjerknes feedback during El Niño and La Niña, possibly coupled with extratropical processes, could affect these patterns.



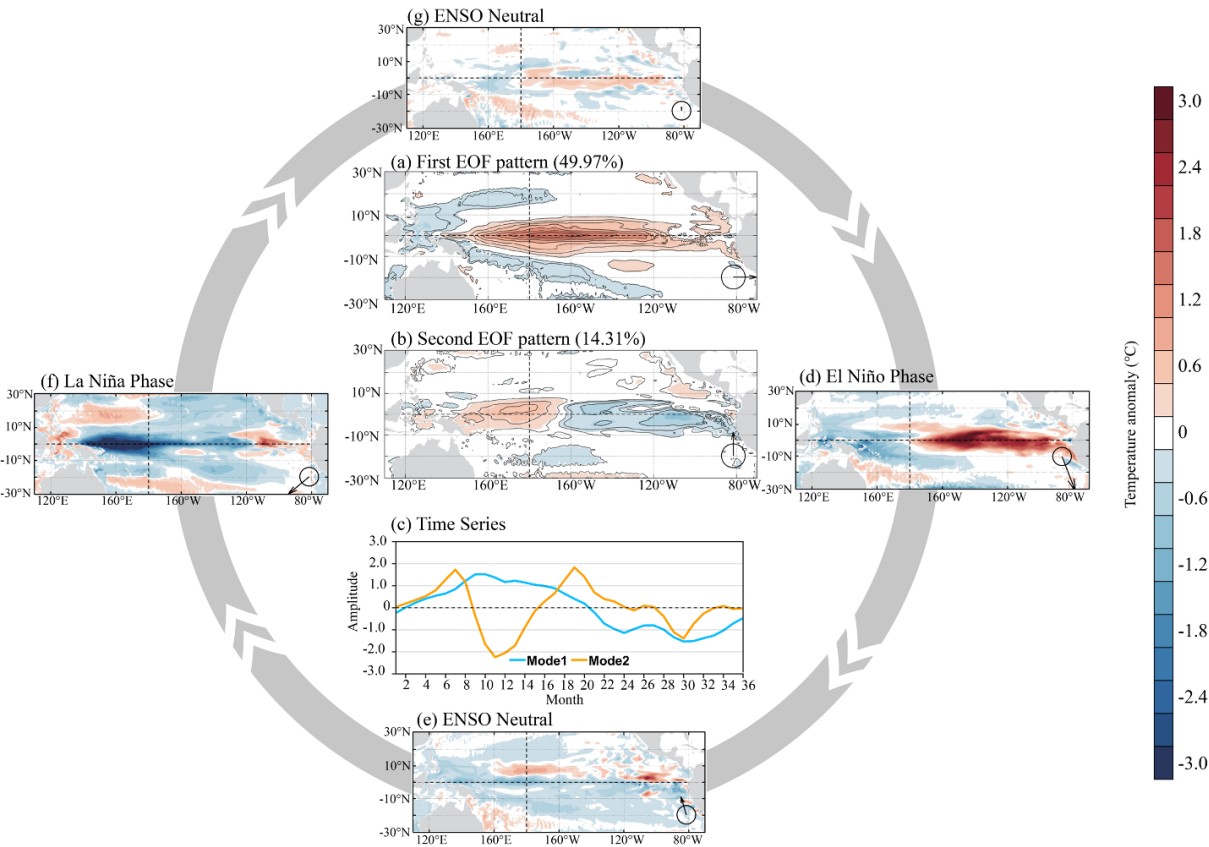

**Figure 11: Distribution of (a) first and (b) second EOF patterns of simulated SSTA in the sensitivity experiment with $\alpha_\tau$=1.5, and (c) their time coefficients. (d-g) are spatial patterns of SSTA over the ENSO cycle. Vectors at the bottom right of (a-b, d-e) show the associated principal components (PCs). The abscissa is PC1, the ordinate is PC2, and the arrow length is the magnitude in PC1–PC2 space (an arrow magnitude of 1 is indicated by the circles).**



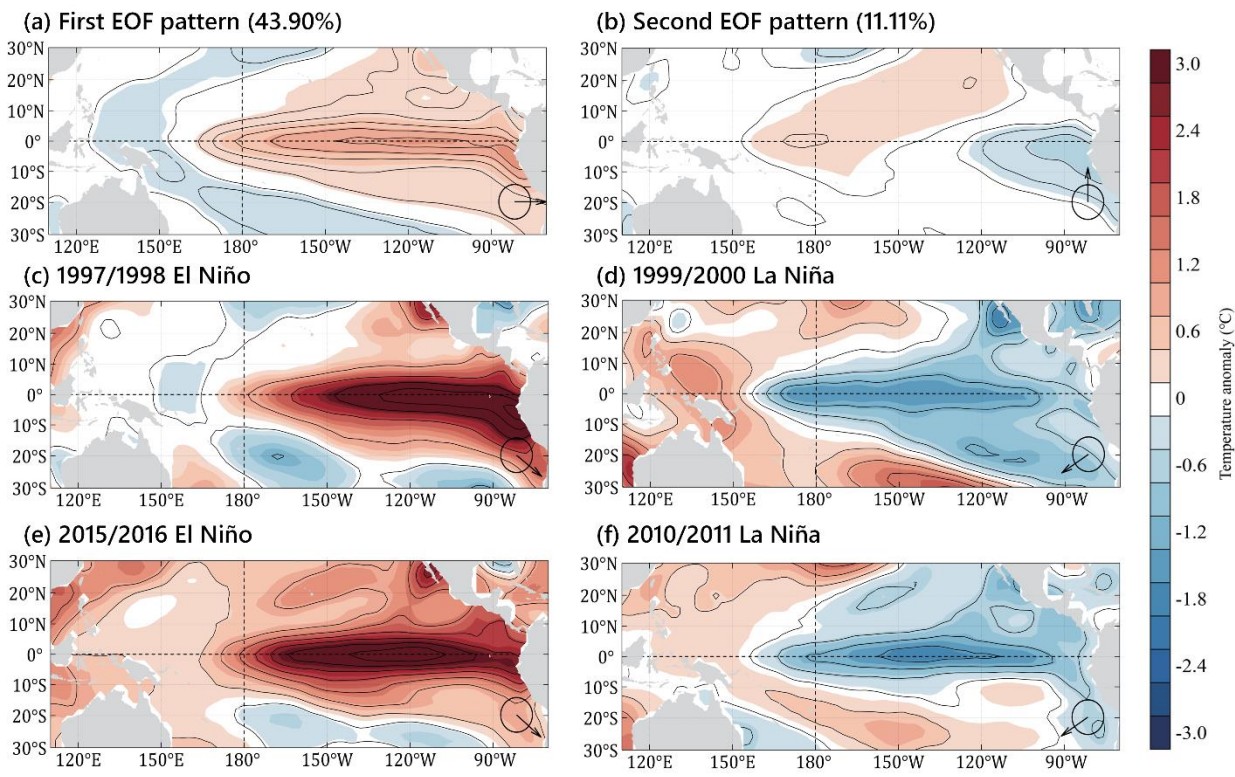

**Figure 12: Distribution of (a) first and (b) second EOF patterns of observed SSTA, and spatial patterns of SSTA in (c) 1997/1998 El Niño, (d) 1999/2000 La Niña, (e) 2015/2016 El Niño, and (f) 2010/2011 La Niña.**

In addition to the simulated interannual SSTA, the HCM$_{ROMS}$ has the advantage of simulating the subsurface temperature changes associated with ENSO. Fig. 13 shows the evolution of 3D temperature anomalies over a complete ENSO cycle, spanning from month 141 to 174, in the sensitivity experiment with $\alpha_\tau$=1.5. It shows that a subsurface warm anomaly forms along the 20 °C isotherm (black contour in Fig. 13) east of the dateline when the simulated El Niño onset (Figs. 13a-b). This subsurface warming intensifies with the simulated El Niño growth (Figs. 13b-c) and reaches its maximum of 3~4 °C, exceeding the surface warming of 2 °C, during the mature stage of the simulated El Niño (Figs. 13d-e). At the same time, a subsurface cooling of -3~-4 °C forms along the 20 °C isotherm west of the dateline (Figs. 13c-e). When the simulated El Niño decays, the subsurface cooling west of the dateline propagates eastward along the 20 °C isotherm, gradually counteracting the subsurface warming in the eastern equatorial Pacific (Figs. 13f-h). While in the simulated onset of La Niña, the subsurface cooling east of the dateline intensifies (Fig. 13i), reaching its peak cold anomaly of -2 °C during the mature stage of the simulated La Niña (Figs. 13j-k). Subsurface warming of 3~4 °C along the 20 °C isotherm west of the dateline emerges during the simulated La Niña (Figs. 13i-k). The subsurface warming west of the dateline offsets the La

Niña-related subsurface cooling in the equatorial eastern Pacific when the simulated La Niña decays (Fig. 13l). The
aforementioned subsurface temperature changes associated with the simulated ENSO in HCM$_{ROMS}$ closely align with the
empirical observations (see Fig. 1 in Timmermann et al. (2018)), validating the HCM$_{ROMS}$ model's ability to depict the
intricate 3D temperature structure changes during the ENSO evolution.

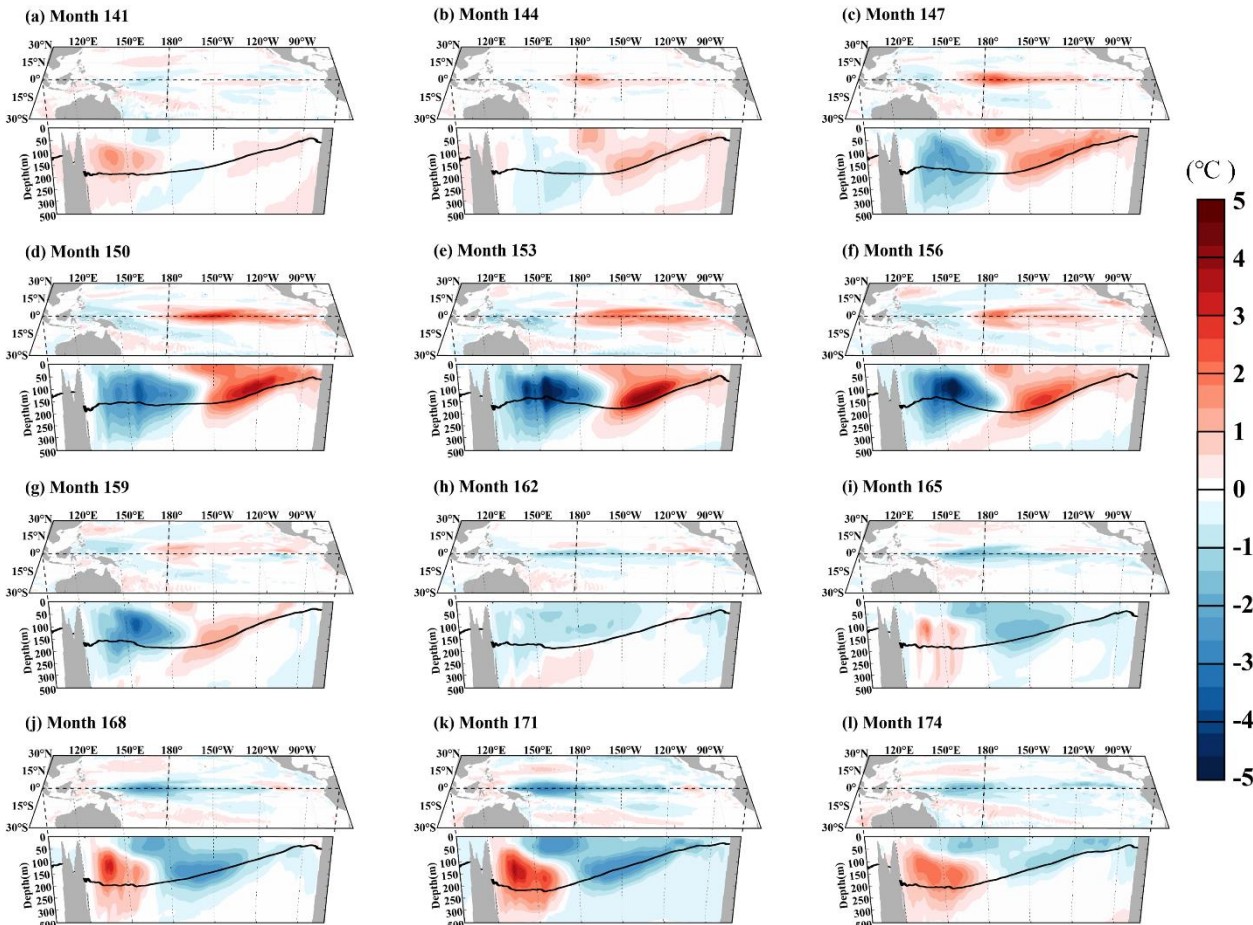


**Figure 13:** Three-dimensional temperature anomalies over the simulated ENSO cycle from month 141 to 174 in the sensitivity
experiment with $\alpha_\tau$=1.5.

**3.4 Budget Analysis on 3D Evolution of the Simulated ENSO**

To better understand the 3D temperature changes related to the simulated ENSO in the HCM$_{ROMS}$, a post-processing heat
budget analysis for the interannual temperature anomaly has been developed in this study. The temperature evolution within
the ROMS model is governed by the advective-diffusive equation:





$$\underbrace{\frac{\partial T}{\partial t}}_{T\_rate} = \underbrace{-\vec{V} \cdot \nabla T}_{T\_adv} + \underbrace{\nabla_h \cdot (K_h \nabla_h T)}_{T\_hdif} + \underbrace{\frac{\partial}{\partial z}\left(K_v \frac{\partial T}{\partial z}\right)}_{T\_vdif} + \underbrace{\frac{Q_R}{\rho_w C_p}}_{T\_solar} \quad ---- (3)$$

where $T$ is the potential temperature; $\vec{V}$ is the current vector; $\nabla$ is the gradient operator; $\nabla_h$ is the horizontal gradient

operator; $K_h$ and $K_v$ are respectively the horizontal and vertical diffusivities; $\rho_w$ is the seawater density; $C_p$ is the specific

heat of seawater at constant pressure; and $Q_R = \frac{\partial F_S}{\partial z}$ is the radiative heating rate, where $F_S$ is the solar radiative heat flux in

the ocean due to solar radiation penetration. The diagnostic terms of $T\_rate, T\_adv, T\_hdif, T\_vdif$, and $T\_solar$ in Eq.3

represent the temperature tendency, total (horizontal + vertical) advection, horizontal diffusion, vertical diffusion, and solar

radiative heating, respectively. We do not separate the total advection effect into the horizontal and vertical components due

to the interconnected nature of these two terms on the monthly time scale, such a separation does not yield additional

interpretability (figs. not shown). The above diagnostic terms are direct outputs of the HCM$_{ROMS}$, which can greatly reduce

the budget error.

We note that the impacts of long-wave radiation and latent and sensible heat fluxes on the temperature changes in the ROMS

model are also included in Eq.3 because these surface heat fluxes serve as the upper boundary condition of the vertical

diffusion term $T\_vdif$ as:

$$K_v \frac{\partial T}{\partial z}\bigg|_{z=\zeta} = \frac{LW + LH + SH}{\rho_w C_p} \quad ---- (4)$$

where $\zeta$ is the sea surface height, $LW$ is the long-wave radiation flux, $LH$ and $SH$ are the latent and sensible heat fluxes

calculated by the bulk heat flux formulas Eq.1 and Eq.2, respectively.

The ENSO-related interannual temperature changes in the HCM$_{ROMS}$ can be diagnosed using the following interannual form

of the budget equation:

$$[T\_rate] = [T\_adv] + [T\_hdif] + [T\_vdif] + [T\_solar] \quad ---- (5)$$

where $[*]$ denotes the interannual operator, which means the simulated values subtract their climatological mean. The

climatological mean of the budget terms utilized in Eq.5 are long-term monthly mean averaged over the last 10 years of the

spin-up. We do not show $[T_{solar}]$ in this study as the solar radiative heat flux in the ocean is calculated by an empirical

irradiance absorption scheme with fixed attenuation depth (see Table. 2 in Paulson and Simpson (1977)) and the solar

radiation forcing at the sea surface is derived from the climatology monthly COREv2 data and thus has no interannual

variability.

During the simulated onset of El Niño (months 141-153; Figs. 13a-e), the $[T\_rate]$ shows a positive temperature tendency of

$2\times10^{-7}$ °C s$^{-1}$ to the east of the dateline, with a maximum warming rate of $3\times10^{-7}$ °C s$^{-1}$ at the depth of 150 m between 150-

120 °W (Fig. 14a), reflecting the 3D temperature changes. The positive $[T\_rate]$ east of the dateline leads to an increase in

the upper ocean temperature, deepening the mean depth of the 20 °C isotherm (red solid line) by 20 m in the eastern Pacific

compared to the simulated climatology (grey solid line; averaged over the same month of the year during the last ten years of

the spin-up). Meanwhile, a negative temperature tendency of -$2\times10^{-7}$ °C s$^{-1}$ is obtained west of the dateline, resulting in



additional cooling and reducing the mean depth of the 20 °C isotherm by 20 m west of the dateline (Fig. 14a). The budget

analysis indicates that the $[T\_rate]$ pattern in Fig. 14a is primarily influenced by the advection effect $[T\_adv]$, wherein the abnormal westerly winds around the dateline (Fig. 10b) during the onset of El Niño induce eastward advection anomalous, leading to the warming (cooling) to the east (west) of the dateline (Fig. 14e). The simulated warming in the subsurface layer exceeding that of the surface during El Niño (Figs. 13a-e) is due to the vertical diffusion effect $[T\_vdif]$. Since the surface heat flux damping during El Niño works to reduce the SST (Fig. 10d), vertical mixing transports the damping effects from

the surface to the deeper layers, partly counteracting the warming in the upper ocean (Fig. 14i).

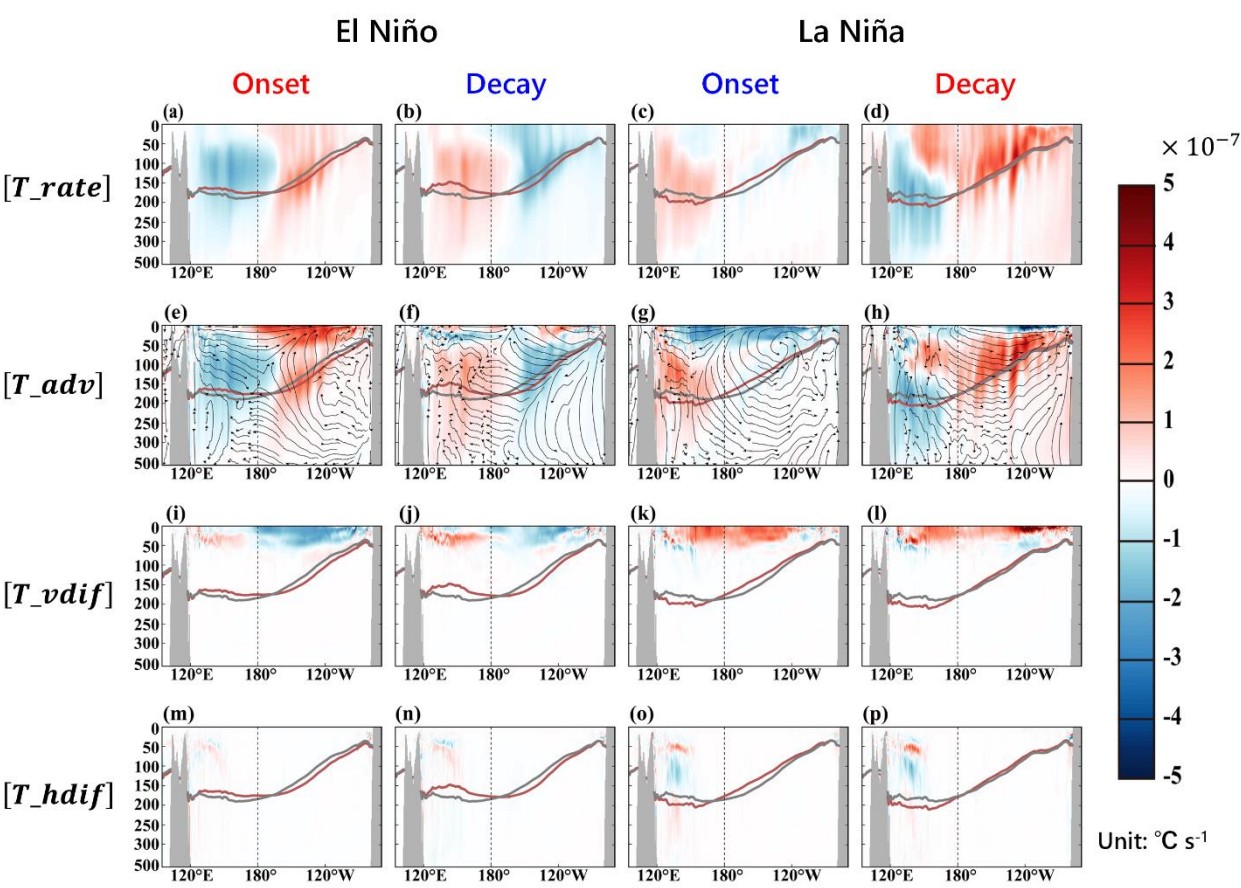

**Figure 14: Vertical cross-section plots of the (a-d) tendency, (e-h) advection, (i-l) vertical diffusion, and (m-p) horizontal diffusion of the interannual temperature budget equation (Eq. 5).**


In the decay phase of the simulated El Niño (months 153-162; Figs. 13e-h), both advection and vertical diffusion effects play constructive roles in shaping the dipole-type temperature changes (Figs. 14b). With the reduction of abnormal westerly





winds during El Niño decay (Fig. 10b), prevailing easterly winds east to the dateline (i.e., trade winds) facilitate westward water transport in the equatorial Pacific. The advection effect $[T\_adv]$ counteracts the accumulated subsurface warming in

the eastern Pacific by transporting warm water to the western Pacific. The descent of warm water around the dateline raises the subsurface temperature in the western Pacific (Fig. 14f). As for the vertical diffusion effect, due to the SST still being warmer than normal during the El Niño decay, the surface heat flux and thus the vertical diffusion effect $[T\_vdif]$ continues to cool the upper ocean (Fig. 10d and 14j).

As for the La Niña period, the budget analysis suggests the advection and vertical diffusion effects play similar roles during

La Niña onset (Figs. 14c, g, and k) and decay (Figs. 14d, h, and l), but with opposite signs, compared to those in the El Niño period. The horizontal diffusion effect $[T\_hdif]$ is not discussed as it only appears in the subsurface west of 150 °E and has a limited impact on ENSO evolution (Figs. 14m-p).

## 4. Conclusions

In this study, we developed a new HCM using the advanced ocean model ROMS (HCM_{ROMS}) to simulate ENSO. Within the

HCM_{ROMS} framework, the ROMS model was coupled with an SVD-based statistical atmospheric model, capturing interannual atmospheric perturbations, including wind stress and FWF anomalies (i.e., $\tau_{inter}$ and $FWF_{inter}$). A turbulent heat flux module, employing bulk approximation and wind-stress-inversed wind speed, was incorporated in the HCM_{ROMS} to address the damping effects of surface heat flux on ENSO dynamics. We executed five sensitivity experiments, adjusting the interannual wind stress coupling coefficient $\alpha_\tau$ ranging from 1.0 to 2.0, to determine the optimal $\alpha_\tau$ for reproducing

sustainable interannual variabilities in the HCM_{ROMS}. A post-processing budget scheme for the interannual temperature anomaly was also developed to analyse processes responsible for ENSO-related 3D temperature changes.

We first examined the performance of the statistical atmospheric model in reproducing the ENSO-related interannual forcing. The retrieved zonal wind stress and FWF anomalies using the statistical atmospheric model forced by the observed SSTA from 1980 to 2020 were compared with observed values from the NCEP/NCAR reanalysis. The SVD-based statistical

atmospheric model replicates the observed anomalous westerly winds originating from the dateline during the major El Niño events of 1982/1983, 1997/1998, and 2015/2016, and the abnormal easterly winds near the dateline in the major La Niña events of 1988/1989, 1998/1999, 1999/2000, 2007/2008, and 2010/2011. The statistical atmospheric model also reproduces the ENSO-related FWF anomaly dipole during El Niño and La Niña. The correlation coefficient between the observed and simulated zonal wind stress anomalies is 0.62 (p<0.01) and the correlation coefficient for the observed and simulated FWF

anomalies is 0.68 (p<0.01).

The ROMS model performance in simulating the mean and seasonal temperatures in the tropical Pacific was also assessed by comparing the simulated climatology with the WOA observations. Although there is a notable overestimation of 1 °C in the mean SST of the western Pacific warm pool during the model spin-up, possibly due to the weaker climatological forcing adopted in the model, the ROMS model replicates the observed SST distribution and the vertical temperature structures





along the equator. The simulated 20 °C isotherm deepens in the western equatorial Pacific, with a maximum depth of 180 m west of the dateline. The simulated 20 °C isotherm depth decreases away from the dateline, reaching a minimum depth of 50 m along the coast of Peru at 90 °W. The ROMS model captures the observed semi-annual SST variation of ±0.5 °C in the western equatorial Pacific and the annual SST variation of ±3 °C in the eastern equatorial Pacific. The semi-annual SST variation in the western equatorial Pacific is attributed to the sun's biannual crossing of the equator while the annual SST

variation in the eastern equatorial Pacific is driven by the effects of trade-wind-induced upwelling.

Sensitivity experiments with different $\alpha_\tau$ values indicate that the optimal $\alpha_\tau$=1.5 can produce sustainable interannual variabilities in the HCM$_{ROMS}$. With the $\alpha_\tau$ set at 1.5, a stable quasi-three-year ENSO cycle, characterized by alternating occurrence of El Niño with a positive SSTA of 2 °C and La Niña with a negative SSTA of -1 °C, exists in the HCM$_{ROMS}$ after the first-eight-month model "initial kick". During the simulated El Niño and La Niña, anomalous westerly wind stress

anomaly of 0.3 dyn cm$^{-2}$ and easterly wind stress anomaly of -0.3 dyn cm$^{-2}$ emerge around the dateline. Surface turbulent heat flux anomaly of ±60 W/m$^2$ appears in the eastern Pacific to dampen the positive and negative SSTAs associated with the simulated El Niño and La Niña, respectively.

EOF analysis was utilized to evaluate the HCM$_{ROMS}$ performance in replicating ENSO-related interannual variability. It revealed that Mode 1 of the simulated SSTA, accounting for 49.97% of the variance, displays a classic El Niño pattern and

demonstrates variability on a quasi-three-year timescale. Mode 2 explaining 14.31% of the variance, depicts a tropical east-west zonal dipole with heightened variability on a 1.5-year timescale. Although the simulated Mode 2 in HCM$_{ROMS}$ fails to capture the influence of the Victoria mode, possibly due to the absence of extratropical processes from the limited model domain of the tropical Pacific. The EOFs of observed and simulated SSTA share certain similarities, including close variance contributions and similar phase configurations. Mode 2 represents the Bjerknes feedback serves as an amplifier

during El Niño but acts as a damper during La Niña. The distinct functions of Mode2 explain the asymmetry between simulated El Niño and La Niña in the HCM$_{ROMS}$.

The HCM$_{ROMS}$ reproduces the 3D temperature changes during ENSO evolution, revealing that the most significant temperature anomalies occur beneath the surface at 150 m. The budget analysis indicates that the interannual temperature tendency $[T\_rate]$ is primarily influenced by the advection effect $[T\_adv]$ driven by interannual wind stress. The vertical

diffusion effect $[T\_vdif]$ also contributes to the formation of subsurface anomaly maxima. However, the primary role of $[T\_vdif]$ is not to amplify cold or warm anomalies within the subsurface layer. Instead, it works to diminish these anomalies within the upper ocean due to the damping effects of turbulent heat flux. Anomalies induced by advection effects in the upper ocean are counteracted by the vertical diffusion effect, resulting in the appearance of subsurface maxima.

The newly developed HCM$_{ROMS}$ provides an effective tool for representing and simulating ENSO in the tropical Pacific

climate system. In this study, we focus on the SSTA-$\tau_{inter}$ coupling, while excluding the FWF effects by setting $\alpha_{FWF}$=0. The SSTA-$FWF_{inter}$ coupling also impacts ENSO and can be flexibly included in the HCM$_{ROMS}$. Further exploration of the FWF effects using the HCM$_{ROMS}$ as a tool will be presented in a future study. In addition, the ROMS model has advantages in modeling multi-scale ocean physical as well biogeochemistry processes, such as mesoscale eddies, tropical instability

waves and ocean biology-induced heating effects. The interactions between multi-scale processes and ENSO will also be a
focal point of future research utilizing the HCM$_{ROMS}$ as a flexible modeling tool.

**Code and data availability**

The Optimum Interpolation Sea Surface Temperature (OISST) data can be downloaded from the National Centers for
Environmental Information interface at https://www.ncei.noaa.gov/products/optimum-interpolation-sst. The SODA3 data
was provided by the UMD Ocean Climate Lab at https://www2.atmos.umd.edu/~ocean/. COREv2 data was provided by
NCAR and is available at https://climatedataguide.ucar.edu/climate-data/corev2-air-sea-surface-fluxes. The ICOADS data is
from the National Oceanic and Atmospheric Administration (NOAA) at https://icoads.noaa.gov/.download.
The HCM$_{ROMS}$ code is available via Zenodo at https://doi.org/10.5281/zenodo.14184175 (Yu, 2024) or GitHub at
https://github.com/clarkyuchina/ROMS-HCM. Interested users can contact the corresponding author for further assistance.

**Author contributions**

Formal analysis: Yang Yu, Yin-nan Li

Writing - original draft: Yang Yu

Writing-review & editing: Yang Yu, Rong-Hua Zhang, Shu-Hua Chen, Yu-Heng Tseng, Wenzhe Zhang, Hongna Wang

**Competing interests**

The contact author has declared that none of the authors has any competing interests.

**Acknowledgements**

This research is supported by the Laoshan Laboratory (No. LSKJ202202402), the National Natural Science Foundation of
China (NSFC; Grant No. 42030410), the Startup Foundation for Introducing Talent of NUIST, and Jiangsu Innovation
Research Group (JSSCTD 202346).

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
