# Peer review of "A Flexible ROMS-based Hybrid Coupled Model for ENSO Studies— Model Formulation and Performance Evaluation"

_Geoscientific Model Development, 2024_

## Referee Comment (RC1)

**Recommendation**: **Accept after Minor Revisions**

**Overall Evaluation**

This article presents a novel Hybrid Coupled Model (HCM$_{ROMS}$) based on the Regional Ocean Modeling System (ROMS) for studying the El Niño-Southern Oscillation (ENSO). The authors provide a detailed description of the model's formulation and evaluate its performance in simulating ENSO-related phenomena. The research is well-conducted, with clear objectives, robust methodology, and comprehensive analysis. The results demonstrate the model's capability to simulate ENSO cycles and associated three-dimensional temperature anomalies, making it a valuable tool for future ENSO research.

1. **P5, Line 108**: "and Um and Vm are calculated time series of left field and left field in the statistical model", the second "left field" should be changed to "right filed"?

2. **P11, Line 228**: We adopt ERSST in this subsection instead of using the NOAA OI SST, "OI SST" => "OISST"

3. **P18, Line 372, 374**: "It shows that the leading EOF (Mode 1)", "The second EOF (Mode 2)". The first and second EOFs have been defined as Mode 1 and Mode 2, respectively, so, please ensure consistency in terminology throughout the subsequent text.

4. **P18, Line 397**: "phase vectors are in the third quadrant" => " the fourth quadrant"

5. **P18, Line 399**: "phase vectors are in the fourth quadrant" => "the third quadrant"

6. **P19, Line 406**: "Vectors at the bottom right of (a-b, d-e) show", "d-e" should be changed to "d-g"?

7. **P22, Line 457**: "where [∗] denotes the interannual operator", the symbol ∗ is not found in Eq. 5

8. **P23, Line 481**: "both advection and vertical diffusion effects play constructive roles in shaping the dipole-type temperature changes (Figs. 14b).", Figs. 14b => Figs. 14b, f, and j

9.  **P23**: The streamlines in Fig 14 need some description.

10. **P24, Line 535**: "The distinct functions of Mode2 explain the asymmetry between",
    Mode2 => Mode 2

---

## Author Comment (AC1)

**Response to Reviewers**
**Manuscript ID gmd-2024-187**

**Responses to the Comments of Reviewer**

*Reviewer #1:*
*Overall Evaluation*
*This article presents a novel Hybrid Coupled Model (HCMROMS) based on the Regional Ocean Modeling System (ROMS) for studying the El Niño-Southern Oscillation (ENSO). The authors provide a detailed description of the model's formulation and evaluate its performance in simulating ENSO-related phenomena. The research is well-conducted, with clear objectives, robust methodology, and comprehensive analysis. The results demonstrate the model's capability to simulate ENSO cycles and associated three-dimensional temperature anomalies, making it a valuable tool for future ENSO research.*

**Our response:** We thank the reviewer for the suggestions (in italic black). We have revised the manuscript based on these suggestions and comments. Below are our point-by-point responses to the reviewer's comments (in blue).

*1. P5, Line 108: "and Um and Vm are calculated time series of left field and left field in the statistical model", the second "left field" should be changed to "right filed"?*
**Our response:** changed

*2. P11, Line 228: We adopt ERSST in this subsection instead of using the NOAA OI SST, "OI SST" => "OISST"*
**Our response:** changed

*3. P18, Line 372, 374: "It shows that the leading EOF (Mode 1)", "The second EOF (Mode 2)". The first and second EOFs have been defined as Mode 1 and Mode 2, respectively, so, please ensure consistency in terminology throughout the subsequent text.*
**Our response:** changed

*4. P18, Line 397: "phase vectors are in the third quadrant" => " the fourth quadrant"*
**Our response:** changed

*5. P18, Line 399: "phase vectors are in the fourth quadrant" => "the third quadrant"*
**Our response:** changed

*6. P19, Line 406: "Vectors at the bottom right of (a-b, d-e) show", "d-e" should be changed to "d-g"?*
**Our response:** changed

*7. P22, Line 457: "where [*] denotes the interannual operator", the symbol *is not found in Eq. 5*
**Our response:** thanks for the comment, we have removed the * in Eq.5

*8. P23, Line 481: "both advection and vertical diffusion effects play constructive roles in shaping the dipole-type temperature changes (Figs. 14b).", Figs. 14b => Figs. 14b, f, and j*
**Our response:** changed

*9. P23: The streamlines in Fig 14 need some description.*
**Our response:** thanks for the comment, we have added a description to it.

*10. P24, Line 535: "The distinct functions of Mode2 explain the asymmetry between", Mode2 => Mode 2*
**Our response:** changed

---

## Author Response (AR1)

**Response to Reviewers**
**Manuscript ID gmd-2024-187**

**REVIEWER 1**

***Reviewer 1, general comment:***
*This article presents a novel Hybrid Coupled Model (HCMROMS) based on the Regional Ocean Modeling System (ROMS) for studying the El Niño-Southern Oscillation (ENSO). The authors provide a detailed description of the model's formulation and evaluate its performance in simulating ENSO-related phenomena. The research is well-conducted, with clear objectives, robust methodology, and comprehensive analysis. The results demonstrate the model's capability to simulate ENSO cycles and associated three-dimensional temperature anomalies, making it a valuable tool for future ENSO research.*

**Our response:** We thank the reviewer for the suggestions (in italic black). We have revised the manuscript based on these suggestions and comments. Below are our point-by-point responses to the reviewer's comments (in blue).

***Reviewer 1, #1****: P5, Line 108: "and Um and Vm are calculated time series of left field and left field in the statistical model", the second "left field" should be changed to "right filed"?*
**Our response:** changed

***Reviewer 1, #2****: P11, Line 228: We adopt ERSST in this subsection instead of using the NOAA OI SST, "OI SST" => "OISST"*
**Our response:** changed

***Reviewer 1, #3****: P18, Line 372, 374: "It shows that the leading EOF (Mode 1)", "The second EOF (Mode 2)". The first and second EOFs have been defined as Mode 1 and Mode 2, respectively, so, please ensure consistency in terminology throughout the subsequent text.*
**Our response:** changed

***Reviewer 1, #4****: P18, Line 397: "phase vectors are in the third quadrant" => " the fourth quadrant"*
**Our response:** changed

***Reviewer 1, #5****: P18, Line 399: "phase vectors are in the fourth quadrant" => "the third quadrant"*
**Our response:** changed

***Reviewer 1, #6****: P19, Line 406: "Vectors at the bottom right of (a-b, d-e) show", "d-e" should be changed to "d-g"?*
**Our response:** changed

***Reviewer 1, #7***: *P22, Line 457: "where [*] denotes the interannual operator", the symbol *is not found in Eq. 5*
**Our response:** thanks for the comment, we have removed the * in Eq.5

***Reviewer 1, #8***: *P23, Line 481: "both advection and vertical diffusion effects play constructive roles in shaping the dipole-type temperature changes (Figs. 14b).", Figs. 14b => Figs. 14b, f, and j*
**Our response:** changed

***Reviewer 1, #9***: *P23: The streamlines in Fig 14 need some description.*
**Our response:** thanks for the comment, we have added a description to it.

***Reviewer 1, #10***: *P24, Line 535: "The distinct functions of Mode2 explain the asymmetry between", Mode2 => Mode 2*
**Our response:** changed

**REVIEWER 2**

*General comment: The work built a hybrid coupled model based on ROMS and a statistical atmosphere. The paper specifically present the model formulation and its performance evaluations about ENSO. Overall, the work is interesting. The developed model will be a useful for future ENSO studies.*
**Our response:** We thank the reviewer for the suggestions (in italic black). We have revised the manuscript based on these suggestions and comments. Below are our point-by-point responses to the reviewer's comments (in blue).

*Reviewer 2, #1: Table 1: I can't understand how your "complexity" is defined, as well as your degree of freedom? The definition of "variables" for dynamical models is different from that for AI. Your rating/table is misleading, and gives one a feeling that the AI models are much more complex than the CGCMs.*
**Our response:** Since 'complexity' is not discussed in the paper and may be misleading, we have removed it from Table 1.

*Reviewer 2, #2: Fig. 1/L121: Not sure how the first SVD modes are derived? SVD needs to be performed by a pair of fields. Please explain how the three fields are used?*
**Our response:** Due to the first SVD mode of $SST_{inter}$ is similar in the $SST_{inter}$-$\tau_{inter}$ and $SST_{inter}$-$FWF_{inter}$ pairings, we only present the result from the $SST_{inter}$-$\tau_{inter}$ pairing in Fig.2a. We have included a description of this in Lines 120–122 as: "We note that since the first SVD mode of $SST_{inter}$ in the $SST_{inter}$-$\tau_{inter}$ and $SST_{inter}$-$FWF_{inter}$ pairings is similar, only the first SVD mode of $SST_{inter}$ from the $SST_{inter}$-$\tau_{inter}$ pairing is shown in Fig. 2a. "

*Reviewer 2, #3: Fig. 9/10: The simulated ENSO cycles are too regular. One suggestion for your future experiments is to add some state-depend noise to your statistical atmospheric model.*
**Our response:** Thanks for the suggestion. Previous studies such as Zhang et al., 2008 have shown that stochastic atmospheric forcing plays an important role in the irregularity of ENSO. In our future work, we will incorporate a stochastic forcing module into the HCM$_{ROMS}$ to reproduce the state-depend noise process.

Reference: Zhang, R.-H., Busalacchi, A. J., and DeWitt, D. G.: The Roles of Atmospheric Stochastic Forcing (SF) and Oceanic Entrainment Temperature (Te) in Decadal Modulation of ENSO, J. Clim., 21, 674–704, https://doi.org/10.1175/2007JCLI1665.1, 2008.

*Reviewer 2, #4: Fig. 14(e-h): Are you presenting streamlines in the figures? If so, are they derived based on mean currents?*
**Our response:** The streamlines are derived from the mean background currents. We have added a description in Fig. 14 as: "The streamlines in (e-h) represent the averaged flow field during the budget calculation."

***Reviewer 2, #5***: *L471: eastward advection anomalous => anomalous eastward advection;*
**Our response:** changed

***Reviewer 2, #6***: *Also, the statements are at least incomplete. The vertical advection of anomalous warm water by mean upwelling is important contributor to the onset of El Ninos. The subsurface warming is also related to the westerly wind anomalies through their triggered downwelling Kelvin waves.*
**Our response:** We agree that the vertical advection of anomalous warm water by mean upwelling is an important contributor to the onset of El Niño. Due to the horizontal and vertical advections are coupled on a monthly time scale and their effects are largely canceled by each other, we did not separate the horizontal and vertical advection individually and only discussed the total advection effect. The effects of downwelling Kelvin waves triggered by westerly wind anomalies are included in the total advection effect. We have address this in Lines 445-446 as:
"The total advection term $T\_adv$ includes both horizontal processes, e.g., the influence of the equatorial currents, and vertical processes, e.g., the downwelling Kelvin waves triggered by the westerly wind anomalies. "

---

## Author Response (AR2)

**Response to Reviewers**
**Manuscript ID gmd-2024-187**

**Responses to the Comments of Reviewer**

*Reviewer #2:*
*This study presents a new hybrid coupled model based on ROMS and a SVD Atmosphere model. The results show that the model can will simulate ENSO cycles and associated three-dimensional temperature anomalies. In the revision, I do see some improvements compared to the previous version. I still have some questions and comments on the details as follows:*
**Our response:** We thank the reviewer for the suggestions (in italic black). We have revised the manuscript based on these suggestions and comments. Below are our point-by-point responses to the reviewer's comments (in blue).

*1. The ROMS model domain is from 95 °E to 70 °W and 30 °S to 30 °N. As we know that ENSO is not only air-sea dynamics of tropical region, extratropical signals also have large influences on ENSO evolutions. Especially for the extreme El Nino events. I wonder how the authors deal with this issue?*
**Our response:** Extratropical processes can significantly influence the evolution of ENSO, particularly during extreme El Niño events. In this study, the model domain is limited to 30°S–30°N and therefore does not include extratropical dynamics. However, since our current focus is on air-sea coupling processes within HCM$_{ROMS}$, we use this domain to isolate and better understand the tropical mechanisms. The role of extratropical influences will be addressed in future work, potentially through improved lateral boundary forcing or an expanded model domain. We address this issue in Lines 147–150 as:

"It should be noted that the limited domain of 30 °S-30 °N excludes extratropical dynamics, which also influences ENSO. The present study primarily focuses on air-sea coupling processes within the tropical Pacific. The role of extratropical influences will be addressed in future work, potentially through enhanced lateral boundary forcing or an expanded model domain."

*2. How does authors dispose boundary conditions of the ROMS model, since they are important for ENSO dynamics?*
**Our response:** The boundary conditions of the ROMS model are derived from the climatological monthly SODA3 data, as mentioned in Lines 210–212: "The climatological monthly SODA3 data also served as the lateral boundary conditions of sea surface height (SSH), currents, temperature, and salinity throughout the model integration."

*3. There are some unclear or long sentences. Such as, Lines 15-20, "For basin-wide applications to the tropical Pacific, here, the ROMS is incorporated with a statistical atmospheric model, which is based on singular value decomposition (SVD), capturing interannual relationships of atmospheric perturbations such as wind stress and freshwater flux anomalies with sea surface temperature (SST) anomalies." Please make them more clearly.*
*Lines 525 "With the $\alpha\tau$ set at 1.5, a stable quasi-three-year ENSO cycle, characterized by alternating occurrence of El Niño with a positive SSTA of 2 °C and La Niña with a negative SSTA of -1 °C, exists in the HCMROMS after the first-eight-month model "initial kick".*

**Our response:** We have rewritten the long sentences as suggested.